# Impact of COVID-19 on Health Emergency and Disaster Risk Management System: A Scoping Review of Healthcare Workforce Management in COVID-19

Odgerel Chimed-Ochir [1,*], Jargalmaa Amarsanaa [1], Nader Ghotbi [2], Yui Yumiya [1], Ryoma Kayano [3], Frank Van Trimpont [4], Virginia Murray [5] and Tatsuhiko Kubo [1]

1 Department of Public Health and Health Policy, Graduate School of Biomedical and Health Sciences, Hiroshima University, Hiroshima 734-8553, Japan; jargalmaa2009@gmail.com (J.A.); yumiya@hiroshima-u.ac.jp (Y.Y.); tkubo@hiroshima-u.ac.jp (T.K.)
2 Graduate School of Asia Pacific Studies, Ritsumeikan Asia Pacific University, Beppu 874-8577, Japan; nader@apu.ac.jp
3 WHO Centre for Health Development, Kobe 651-0073, Japan; kayanor@who.int
4 European Council of Disaster Medicine, B5500 Dinant, Belgium; frank.van.trimpont@gmail.com
5 Global Disaster Risk Reduction, UK Health Security Agency, London SW1P 3JR, UK; virginia.murray@ukhsa.gov.uk
* Correspondence: odgerel@hiroshima-u.ac.jp

**Abstract:** During the COVID-19 pandemic, many countries faced a shortage as well as maldistribution of healthcare workers and a misalignment between healthcare needs and worker skills. In this scoping review, we have sought to identify the country-level responses to health workforce shortages during the COVID-19 pandemic as well as the advantages/best practices and disadvantages/lessons learned. We have reviewed 24 scientific papers in four electronic databases: Medline, Web of Science, CINAHL, and TRIP. The main strategies implemented by countries were financial coordination mechanisms, relaxing standards/rules, worker redeployment, recruitment of volunteers, fast-tracking medical students, and using other workforce resources such as the recruitment of inactive healthcare workers and returnees whose registration had lapsed within the preceding 1–2 years. These strategies demonstrated numerous advantages, such as establishing mutual support across nations and organizations, boosting motivation among healthcare workers, and creating a new staffing model for future pandemics. However, several important lessons were also learned during the implementation process. Managing volunteers, including ensuring their safety and allocating them to areas in need, required significant effort and high-level coordination, particularly in the absence of a comprehensive needs assessment.

**Keywords:** COVID-19; health-EDRM; health workforce; pandemic response; scoping review

## 1. Introduction

The first case of COVID-19 was identified in Wuhan (China) in December 2019 [1] and, as of 6 April 2023, approximately 762 million individuals had been infected globally and more than 6.9 million had died with the disease [2]. The COVID-19 pandemic conforms to key baseline characteristics of a disaster, which is defined as "a situation or event that overwhelms local capacity, necessitating a request for national or international level of assistance" [3].

The COVID-19 pandemic severely disrupted the health sector's human resources and health infrastructure, particularly in countries with chronic labor shortages and/or inadequate skill-mix profiles. WHO stated that "there is no health without a health workforce" [4]. This was particularly noticeable during the pandemic [5], when the health workforce emerged as a fundamental part of how countries worked to provide health services [6–8]. The WHO's Health Emergency and Disaster Risk Management (Health-EDRM)

Framework, which sought to guide countries on developing capacities to reduce the risks and impacts of all types of emergencies and disasters, including epidemics and pandemics, stated that human resources are one of the ten functions that need to be maintained to improve countries' resilience to disaster (Scheme 1). In a Health-EDRM setting, the key human resource management considerations include planning for staffing requirements, including surge capacity for emergency response, and education and training for competency development, occupational health, and occupational safety. The current scoping review will focus on the strategies used during the COVID-19 pandemic in planning for staffing requirements.

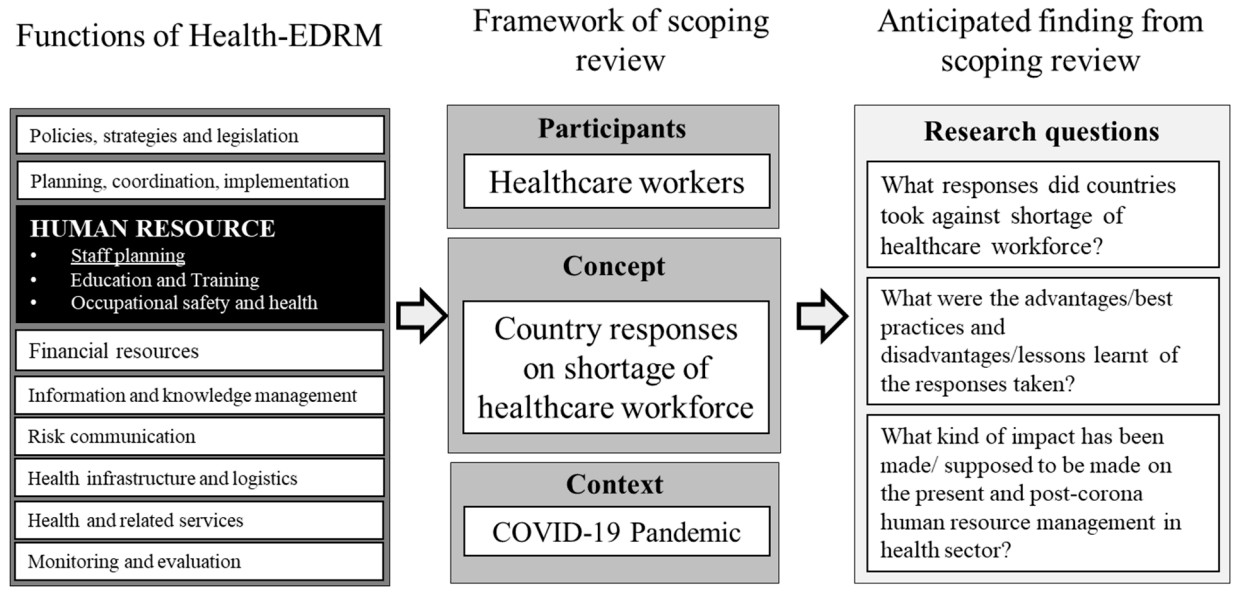

**Scheme 1.** Logic model of the research.

The WHO collaborated with the Asia Pacific Observatory and the European Observatory on Health Systems and Policies to publish the COVID-19 Health System Response Monitor, which contained information on how the health systems of European and some Asian countries responded to the pandemic, including responses to shortages in the healthcare workforce [9,10]. However, these reports covered only the first and second wave of the pandemic (i.e., 2020 and early 2021, respectively); new strategies may have emerged in subsequent waves and/or more countries may have implemented certain strategies; the reports lacked information on some activities in some countries, including Germany [11], and the reports did not cover the advantages/best practices and disadvantages/lessons learned from the implemented strategies. In this scoping review, our objective was to thoroughly explore and identify the various responses and strategies implemented at the country level to tackle the challenge of healthcare workforce shortages. Moreover, we sought to go beyond a mere identification of responses and delve deeper into understanding the underlying factors that contribute to their effectiveness or limitations. In doing so, we aimed to capture not only the advantages and best practices associated with these responses but also the potential disadvantages and valuable lessons learned from their implementation. Such information offers valuable insights and recommendations that may help inform policy decisions, guide future research, and ultimately contribute to the development of more effective and sustainable solutions in addressing healthcare workforce shortages at a global scale during any similar crisis in the future.

## 2. Materials and Methods

The design of the scoping review framework is based on the Joanna Briggs Institute (JBI) Framework of evidence synthesis, which consists of five stages [12]:

1. Identifying the research question;
2. Identifying relevant studies;
3. Selecting studies;
4. Presenting data;
5. Collating results.

### 2.1. Identifying the Research Question

As indicated by the JBI Framework of evidence synthesis, the review question included information on the "participants", the main focus or "concept", and the "context" of the review (PCC—Participant, Concept, Context). In the current review, "participants" are healthcare workers (healthcare workforce), the "concept" is the country's responses to workforce shortage, and the "context" is the COVID-19 pandemic. Based on the defined PCC framework, the following research questions were developed:

- What responses were made by countries against shortages in the healthcare workforce?
- What were the advantages/best practices of the responses taken?
- What were the disadvantages/lessons learnt of the responses taken?

### 2.2. Identifying Relevant Studies

Search strategy: we searched for relevant scientific papers in four electronic databases: Medline, Web of Science, CINAHL, and TRIP. The strategy and key terms used in the search were: "Healthcare personnel" AND "Healthcare workforce" AND "COVID-19". The complete search strategy is presented in Table 1.

**Table 1.** Search strategy of scoping review.

| Database | Search Term | Number of Articles |
|---|---|---|
| PubMed * | (("COVID-19/epidemiology"[MeSH Terms] OR "COVID-19/organization and administration"[MeSH Terms]) AND ("workforce/economics"[MeSH Terms] OR "workforce/organization and administration"[MeSH Terms] OR "workforce/supply and distribution"[MeSH Terms] OR "surge capacity/organization and administration"[MeSH Terms] OR ("personnel management/adverse effects"[MeSH Terms] OR "personnel management/instrumentation"[MeSH Terms] OR "personnel management/methods"[MeSH Terms] OR "personnel management/organization and administration"[MeSH Terms] OR "personnel management/psychology"[MeSH Terms] OR "personnel management/statistics and numerical data"[MeSH Terms] OR "personnel management/supply and distribution"[MeSH Terms]))) Filters: Free full text, English | 251 |
| Web of Science | COVID-19 (Topic) and "health* workforce" OR "human resource" (Topic) and Public Environmental Occupational Health or Health Policy Services or Multidisciplinary Sciences (Web of Science Categories) and English (Languages) | 293 |
| CINAHL | TX healthcare workforce shortage OR TX human resource management AND TX COVID-19 Limiters—Linked Full Text; Published Date: 20200101–20221231 Expanders—Apply equivalent subjects Narrow by Language:—English Search modes—Boolean/Phrase | 46 |
| TRIP Medical Database | PICO: Population (healthcare workers OR doctors OR nurses) AND Intervention (COVID-19) AND Outcome: (country responses OR intervention OR strategies) | 12 |

\* Mesh "Health workforce" is a subcategory of Mesh "Workforce". Therefore, we selected Mesh "Workforce" to capture a broader scope of articles.

Table 2 depicts the inclusion and exclusion criteria for scoping review.

**Table 2.** Inclusion and exclusion criteria for scoping review.

| | Inclusion Criteria | Exclusion Criteria |
|---|---|---|
| Population | Healthcare workers (including physicians, nurses, allied health professionals, and support staff) working in any healthcare setting (including hospitals, primary care clinics, and community health centers) during the COVID-19 pandemic | Non-healthcare workers |
| Concept | Any type of responses that were taken to support the shortage of healthcare workforce at any levels including organizational, local, regional, and international | Only highlighting the problems and challenges of healthcare workforce but not addressing the response to these issues |
| Context | COVID-19 pandemic, including any phase of the pandemic (e.g., initial outbreak, subsequent waves, vaccination campaigns) | Non-COVID-19 infection including SARS, MERS |
| Publication type | All types of publications that were related to responses for healthcare workforce shortages during the COVID-19 pandemic and published in peer-reviewed journals | Articles in pre-print server, other publication venues, including books, book chapters and gray literature |
| Publication date | 1 January 2020 to 8 August 2022 | NA |
| Study design | Any study design, including empirical and non-empirical works that answer the research questions | Clinical randomized trials |
| Language | English | Other than English |

### 2.3. Selecting Studies

After removal of duplicates from the initial search result, a primary screening by title and abstract was conducted independently by two researchers (OC and JA). Papers with no consensus were included for full-text review. The full text of articles deemed relevant for inclusion were screened. The relevant references that were cited in articles selected from the full-text review were also reviewed to identify additional relevant articles. The search for articles was visualized using the Preferred Reporting Items for Systematic Reviews and Meta-Analysis (PRISMA) flow chart [13].

### 2.4. Presenting Data

OC and AJ extracted the following information from articles: authors, title of article, publication date, period covered by the study, study design, country, types of responses taken against health workforce shortages, and advantages/best practices and disadvantages/lessons learned from these responses. Data from the included articles were extracted to an Excel spreadsheet pre-populated with the above-listed items by OC and AJ. When agreement could not be reached between the researchers (OC, AJ), a third researcher (TK) adjudicated at this stage.

### 2.5. Collating Results

For all included studies, a summary of basic bibliometric information was presented. Qualitative narrative synthesis was undertaken in accordance with the pre-populated items.

## 3. Results

### 3.1. General Description of Reviewed Articles

A total of 602 articles were retrieved from the four databases. After removal of duplicates and screening by title and abstract, 67 articles were retained for full-text review. After the full-text review, 12 articles remained for analysis. After the relevant references cited in those 12 papers were also screened, a final total of 18 articles were analyzed in the study (Figure 1).

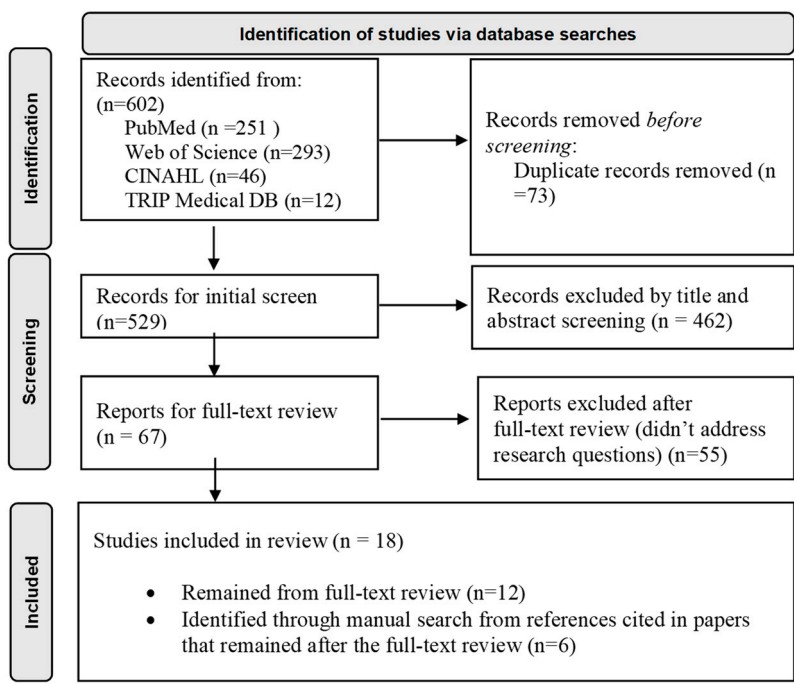

**Figure 1.** PRISMA flow chart of the study selection for the scoping review.

To accurately determine the timeframe of the responses, we opted to utilize the study period instead of the publication year because of the potential time lag between the period a study covered and its publication date, which can often extend from several months to a year. Of the 18 scientific papers analyzed, 61% covered information from the year 2020 while 39% covered the year 2021. No included article covered the year 2022. The reviewed papers consisted of commentaries, including summaries, case reports, and policy and practices (28% of the included papers), followed by reviews (28%), research articles (22%) and special reports (17%). (Figure 2).

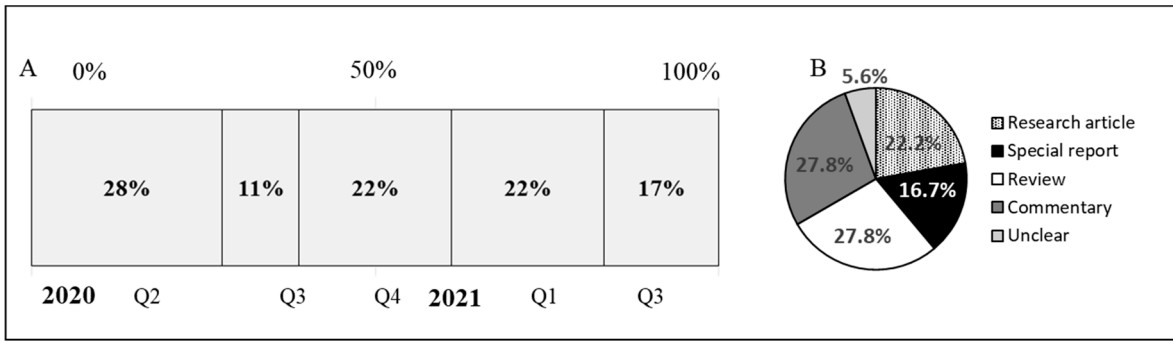

**Figure 2.** Bibliometric information on the reviewed articles. (**A**) Distribution of reviewed papers by study period covered (Q- Quarter). (**B**) Types of papers reviewed and their proportions.

Detailed information on the reviewed articles is presented in Table 3.

**Table 3.** Information on articles used in scoping review.

| N | Author Information | Article Title | Journal Published | Article Type | Study Covering Period | Publication Year | Country |
|---|---|---|---|---|---|---|---|
| 1 | Burau V, Falkenbach M, Neri S, Peckham S, Wallenburg I, Kuhlmann E. [5] | Health system resilience and health workforce capacities: Comparing health system responses during the COVID-19 pandemic in six European countries. | The International Journal of Health Planning and Management | Research article | June 2020 | 2022 | England, Germany, Denmark, Netherlands, Denmark, Austria |
| 2 | Köppen J, Hartl K, Maier CB. [11] | Health workforce response to COVID-19: What pandemic preparedness planning and action at the federal and state levels in Germany? Germany's health workforce responses to COVID-19. | The International Journal of Health Planning and Management | Research article | May 2020 | 2021 | Germany |
| 3 | Carroll WD, Strenger V, Eber E, et al. [14] | European and United Kingdom COVID-19 pandemic experience: The same but different. | Paediatric Respiratory Reviews | Review | June 2020 | 2020 | Italy, UK |
| 4 | Rees GH, Peralta Quispe F, Scotter C. [15] | The implications of COVID-19 for health workforce planning and policy: the case of Peru. | The International Journal of Health Planning and Management | Special report | December 2020 | 2021 | Peru |
| 5 | Dinić M, Šantrić Milićević M, Mandić-Rajčević S, Tripković K. [16] | Health workforce management in the context of the COVID-19 pandemic: A survey of physicians in Serbia. | The International Journal of Health Planning and Management | Article | December 2020 | 2021 | Serbia |
| 6 | Said D, Brinkwirth S, Taylor A, Markwart R, Eckmanns T. [17] | The Containment Scouts: First Insights into an Initiative to Increase the Public Health Workforce for Contact Tracing during the COVID-19 Pandemic in Germany. | International Journal of Environmental Research and Public Health | Project report | July 2021 | 2021 | Germany |
| 7 | Waitzberg R, Hernández-Quevedo C, Bernal-Delgado E, et al. [18] | Early health system responses to the COVID-19 pandemic in Mediterranean countries: A tale of successes and challenges [published correction appears in Health Policy]. | Health Policy | Review | October 2021 | 2022 | Cyprus, Greece Israel, Italy, Malta, Portugal, Spain |
| 8 | Nittayasoot N, Suphanchaimat R, Namwat C, Dejburum P, Tangcharoensathien V. [19] | Public health policies and health-care workers' response to the COVID-19 pandemic, Thailand. | The Bulletin of the World Health Organization | Policy & Practice | August 2020 | 2021 | Thailand |
| 9 | Webb E, Winkelmann J, Scarpetti G, et al. [20] | Lessons learned from the Baltic countries' response to the first wave of COVID-19. | Health Policy | Summary | March 2021 | 2021 | Latvia, Estonia, Lithuania |
| 10 | Winkelmann J, Webb E, Williams GA, Hernández-Quevedo C, Maier CB, Panteli D. [21] | European countries' responses in ensuring sufficient physical infrastructure and workforce capacity during the first COVID-19 wave. | Health Policy | Comprehensive review | February 2021 | 2022 | Albania, Armenia, Austria, Belgium, Bosnia and Herzegovina, Bulgaria, Croatia, Cyprus, Denmark, England, Estonia, France, Germany, Hungary, Iceland, Ireland, Italy, Germany, Lithuania, Luxembourg, Malta, Montenegro, Monaco, Netherlands, Norway, North Macedonia, Poland, Portugal, Romania, Russia, San Marino, Romania, Serbia, Slovenia, Spain, Sweden, Switzerland, Portugal, Turkey, Ukraine |

**Table 3.** *Cont.*

| N | Author Information | Article Title | Journal Published | Article Type | Study Covering Period | Publication Year | Country |
|---|---|---|---|---|---|---|---|
| 11 | Bourgeault IL, Maier CB, Dieleman M, et al. [22] | The COVID-19 pandemic presents an opportunity to develop more sustainable health workforces. | Human Resources for Health | Commentary | December 2020 | 2020 | Netherlands, Germany, Australia, Jamaica, UK, Canada, Mexico, China, USA |
| 12 | Muhammad Nur Amir AR, Binti Amer Nordin A, Lim YC, Binti Ahmad Shauki NI, Binti Ibrahim NH. [23] | Workforce Mobilization From the National Institutes of Health for the Ministry of Health Malaysia: A COVID-19 Pandemic Response. | Frontiers in Public Health | Community case study | June 2020 | 2021 | Malaysia |
| 13 | Divito M, Advincula A, Burgansky A, et al. [24] | Intradepartmental redeployment of faculty and staff. | Seminars in Perinatology | Unclear | July 2020 | 2020 | USA |
| 14 | Zhu P, Liu X, Wu Q, Loke J, Lim D, Xu H. [25] | China's Successful Recruitment of Healthcare Professionals to the Worst-Hit City: A Lesson Learned. | International Journal of Environmental Research and Public Health | Article | July 2021 | 2021 | China |
| 15 | Collins GB, Ahluwalia N, Arrol L, et al. [26] | Lessons in cognitive unloading, skills mixing, flattened hierarchy and organisational agility from the Nightingale Hospital London during the first wave of the SARS-CoV-2 pandemic. | BMJ Open Quality | Narrative review | February 2021 | 2021 | England |
| 16 | Vera San Juan N, Clark SE, Camilleri M, et al. [27] | Training and redeployment of healthcare workers to intensive care units (ICUs) during the COVID-19 pandemic: a systematic review. | BMJ Open | Review | February 2021 | 2022 | |
| 17 | Satterfield CA, Goodman ML, Keiser P, et al. [28] | Rapid Development, Training, and Implementation of a Remote Health Profession's Student Volunteer Corps During the COVID-19 Pandemic. | Public Health Reports | Case Reports/Practice | August 2021 | 2021 | USA |
| 18 | Bahethi RR, Liu BY, Asriel B, et al. [29] | The COVID-19 Student WorkForce at the Icahn School of Medicine at Mount Sinai: A Model for Rapid Response in Emergency Preparedness. | Academic Medicine | Innovation Report | December 2020 | 2021 | USA |

Table 4 summarizes the country-level responses of countries, and their advantages/best practices and disadvantages/lessons learned, as captured in our review. More detailed information is presented in Supplementary Table S1.

**Table 4.** Summary of study findings.

| Response Examples | Advantages/Best Practices | Disadvantages/Lessons Learned |
|---|---|---|
| Financial coordination mechanism | | |
| ➢ Governments allocated additional budget to hire new staff, contract workers involved in clinical tasks, change part-time contracts to full-time contracts, and make new agreements with wage increases. <br> ➢ Governments/institutions provided financial support for HCWs (reimbursement of healthcare-related expenses, and provision of medical insurance, salary increases, and paid leave). <br> ➢ Governments provided hospitals with financial compensation for unoccupied beds to enable HCWs to focus on COVID-19-related tasks. <br> ➢ Other financial incentives to enable non-medical personnel and students to become involved in COVID-19 tasks. | ➢ Financial support worked to motivate HCWs. <br> ➢ When bed occupancy rates decreased, staff could focus more on patients with COVID-19. <br> ➢ Providing a salary and contracts of at least 6 months to containment scouts (i.e., medical students and non-medical personnel) was found to yield longer-term support than other volunteer-based programs. | ➢ If a pandemic or other disaster lasts for an extended period, countries may face a financial burden because most of these strategies were intended to be only short-term. |
| Relaxing standards and rules | | |
| ➢ Relaxing quality standards: The Netherlands allowed pharmacists to prepare the drips at pharmacies instead of distributing medicines separately. <br> ➢ Extending working hours: workers were allowed to work longer and back-to-back shifts. <br> ➢ Loosening of hiring requirements to support fast recruitment. | ➢ Relaxing quality standards resulted in time savings for ICU nurses. | ➢ Extension of working hours led to burnout among HCWs. |

**Table 4.** *Cont.*

| Response Examples | Advantages/Best Practices | Disadvantages/Lessons Learned |
|---|---|---|
| Redeployment/task shifting/skill mixing | | |
| International level:<br>➢ Some countries sent their medical teams to other countries experiencing the heights of their infection surges.<br><br>National level:<br>➢ Governments organized redeployments from public or private hospitals to public hospitals.<br>➢ Mandated graduating doctors to deal with staff shortages in rural areas.<br><br>Subnational level:<br>➢ States, regions, and provinces authorized/organized mutual support systems to redeploy their HCWs to critical areas. | General/system level:<br>➢ In many countries, emergency legislation paved the way for various approaches to rapidly mobilize and recruit health workers.<br>➢ Authorization of mutual support systems enabled HCWs to work across state lines and in regions with high care needs.<br>➢ Redeployment built solidarity to support regions and countries requiring more surge capacity. Collaboration at national/subnational levels was useful in decreasing hospital overcrowding and supporting health services in regions where services were not available.<br>➢ Redeployment supported workforce flexibility (i.e., flexibility in staffing ratios). Some countries relaxed their rules on HCW numbers and allowed more flexibility in hospital placements of nurses.<br>➢ Training up more physically available staff proved to be good practice. For example, Canada trained up more registered nurses to handle ventilators in case there was a shortage of respiratory therapists, because there were many more registered nurses than respiratory therapists in the jurisdiction.<br>➢ Peru produced a baseline dataset to identify skill needs and design more appropriate models of redeployment, and thus now has an available store of workforce intelligence on placement choices and role preferences.<br>➢ Task shifting and new skill-mixing innovations leveraged the full scope of skills available within and outside of the health workforce. | General/system level:<br>➢ The implementation of many of these strategies necessitated the adoption of emergency legislation.<br>➢ There were some conflicts between central governments and regions, which made coordination difficult.<br>➢ Jurisdictions often lack some basic guidelines on workforce adequacy, such as the required staffing ratios of physicians and nurses during "normal" and "emergency" situations. This hindered efforts to monitor and adjust resources during the pandemic.<br>➢ Baseline data on HCWs' qualifications, specialties, availability, and role preferences were unavailable when redeployment was needed during the emergency. The lack of a medical volunteer registry to recruit personnel slowed initial deployments. |

**Table 4.** *Cont.*

| Response Examples | Advantages/Best Practices | Disadvantages/Lessons Learned |
| --- | --- | --- |
| | Organizational level: | |
| | ➢ Redeployment enabled exchanges and mutual help between medical teams, such as in sharing protective materials, discussing cases, and exchanging treatment experiences. | Organization level: |
| | | ➢ Short-term and intensive trainings were needed to upskill new recruits. |
| | ➢ Good leadership helped to facilitate the successful redeployment of medical teams and build temporary teams. For example, leaders allocated human resources based on their expertise and this enabled mutual support, case discussions, and experience sharing. | ➢ For external and internal redeployment, needs assessments were crucial for avoiding unnecessary use of human resources. |
| Organizational level: | ➢ Decentralized leadership approaches better facilitated targeted training, as local leaders were better able to identify the training needs of each redeployee. | ➢ It was crucial to maximize the use of each HCW's experience and current knowledge by placing them in roles where existing skills could be more easily transferrable. |
| ➢ Hospitals reorganized units providing non-essential services to provide COVID-19 services. Elective procedures were canceled. Doctors not involved in COVID-19 services were redeployed to areas in need. | ➢ Redeploying HCWs to designated treatment teams (task-based units made up of multidisciplinary teams with clear leadership and constant communication) was a successful strategy. The members of treatment teams were assigned to complete a specific necessary step of intensive care when requested by experienced ICU HCWs. | ➢ A key barrier for successful redeployment planning was difficulties in measuring the need for human resources (i.e., identifying which specific roles were in demand and which members of the workforce were available and healthy). |
| Implemented tiered staffing models in which critical care physicians or nurses oversaw non-ICU clinicians. For example, experienced renal physicians, together with trainee radiologists, developed line insertion teams, and orthopedists and physiotherapists assisted with proning. | ➢ The tiered model represented an important shift in ways of working and understanding collaborations between health specialists. The model reduced the personnel required for procedures, reduced aerosolization of the virus, decreased the time dedicated to procedures, and required little or no training for personnel to provide assistance. | ➢ There were concerns about deployed healthcare personnel's skills and competencies. |
| | Personnel level: | Personnel level: |
| | ➢ Some HCWs perceived redeployment to be a rewarding experience. | ➢ Changes in schedules, workloads, working hours, and workplaces brought swift and significant disruptions to doctors' working lives. |
| | ➢ There was greater recognition and acknowledgement of public health doctors at both organizational and policy levels. | ➢ Gaps existed between expectations and perceptions for duties. |
| | ➢ Any type of incentive (free transport, issuance of credit hours) helped encourage redeployees who were working in unpopular conditions, such as weekend and night shifts. | ➢ HCWs strongly feared COVID-19 exposure. |
| | Feedback from redeployed staff was helpful for other redeployed staff. | HCWs had concerns relating to their skills, patient safety, and professional losses. |

**Table 4.** *Cont.*

| Response Examples | Advantages/Best Practices | Disadvantages/Lessons Learned |
|---|---|---|
| | Recruiting volunteer/fast-tracking medical students | |
| ➢ Non-medical licensed volunteers were recruited to support public health responses.<br>➢ Medical students were recruited to support childcare for HCWs to public health activities.<br>➢ Medical students were fast tracked, enabling them to contribute to COVID-19 responses. | <u>General:</u><br>➢ Volunteering led to a new staffing model that can be used in a future pandemic. Through volunteer activities and task shifting, other diseases and non-severe cases could be managed through community settings and home care.<br>➢ Many public health tasks can be managed by volunteers, allowing HCWs to focus on tasks that require specialization.<br>➢ If a scouting or volunteer program is coordinated by a statutory body and integrated into the statutory public health system, it can be implemented nationwide. This contrasts with programs that are limited to local or regional areas and not integrated into a national strategy.<br>➢ By providing a salary and contract for a certain period, an initiative can offer longer-term support than other volunteer-based programs.<br>➢ Timely information sharing among volunteers about their tasks and performances was crucial to learn from each other. A survey and video of volunteer experience was produced which enabled volunteers to learn from each other. Training materials for volunteers reflecting updates on recommendations for contact tracing and contact management were made available at the website of the Robert Koch Institute. | <u>General:</u><br>➢ Identifying volunteers was challenging (time-consuming, labor-intensive) in the absence of a registry system.<br>➢ Carefully established data management and relevant information availability could facilitate the recruitment of volunteers.<br>➢ There was some concern about the qualifications of volunteers. Of them, 20% had qualifications evaluated as "insufficient" by local health authorities. |

**Table 4.** *Cont.*

| Response Examples | Advantages/Best Practices | Disadvantages/Lessons Learned |
|---|---|---|
| ➢ Non-medical licensed volunteers were recruited to support public health responses. <br> ➢ Medical students were recruited to support childcare for HCWs to public health activities. <br> ➢ Medical students were fast tracked, enabling them to contribute to COVID-19 responses. | Organizational level: <br><br> ➢ Involvement of skilled volunteers enabled frontline workers to focus on work requiring non-transferable skills. <br> ➢ Bidirectional engagement is crucial for volunteer activity. For example, school administrations agreed to use volunteer hours for scholarship requirements. <br> ➢ Well-organized approaches for recruiting and managing volunteers needed to be established. For example, in Mount Sinai School, a Student Council Emergency Preparedness Committee was created to build a labor force of medical students and nursing candidates. They were distributed into task groups (e.g., pharmacy, administrative services, hospital operations, labs/research, personal protective equipment, telehealth, and morale). When the committee received a detailed request from a department, the head of each task group assigned available students to the field. <br> ➢ Flexibility was a key. Students could report on any concern via a volunteer log or email the staff in charge. There were no strict work hours; overwhelmed students could regulate their own time. <br> ➢ Village health volunteers who shared the dialect, religion, and sociocultural practices of local communities were invaluable in challenging circumstances. | Organizational level: <br><br> ➢ Management of volunteers, including supervision of safety and allocation to areas in need, required great deals of effort and coordination, especially in the absence of a needs assessment. <br> ➢ An overwhelming number of volunteers were enrolled following the inception of the NIH COVID-19 operation room. The screening of volunteers was time-consuming and the calling up of identified volunteers was labor-intensive. |
| | Personnel level: <br><br> ➢ Volunteer activity was made an eligibility criterion for scholarships, and students could use volunteer hours to meet scholarship requirements. | Personnel level: <br><br> ➢ Managers were concerned about the skill levels and competencies of fast-tracked medical students and their potential impacts on patient safety. |

Table 4. *Cont.*

| Response Examples | Advantages/Best Practices | Disadvantages/Lessons Learned |
| --- | --- | --- |
| | Using other workforce resources | |
| ➢ Recalled inactive healthcare workers. <br> ➢ Recruited returnees whose registration has lapsed within the prior 1–2 years. <br> ➢ Integrated internationally educated health professionals. | General: <br> ➢ Leveraged the full scope of skills available outside the health workforce. <br> ➢ An NGO set up a database of inactive health workers who could volunteer in a case of need, and hospitals worked together to reassign staff with COVID-19 training. | General: <br> ➢ An official reserve list with available health professionals could be helpful. <br> ➢ The calling up of internationally educated HCWs neglected local underemployment. <br> ➢ When calling internationally educated HCWs, there was lack of clarity regarding contract terms, visas, workers' skills, and competency. <br><br> Organizational level: <br> ➢ Short-term training was still needed. <br> ➢ Politicians were concerned about safety implications. <br><br> Personnel level: <br> ➢ There were greater potential virus exposure/infection risks among older workers. |

### 3.2. Financial Coordination Mechanisms

Hiring new staff: the governments of some of the studied countries allocated additional budget resources to augment medical and nursing staff through the hiring of new staff [14–18]. For example, during the period that Serbia was under a state of emergency because of the pandemic, the Minister of Health of Serbia announced the recruitment of 1500 physicians [16] and the additional deployment of 200 health workers for a period of 6 months, using European Union funds amounting to approximately EUR 1 million [16]. These workers primarily helped with detecting COVID-19 cases in public health institutes. The German Federal Ministry of Health implemented a project to improve the public health workforce capacity during the period from March 2020 to the end of 2020 and spent approximately EUR 11.3 million for the first phase. Germany also changed part-time contracts for healthcare workers (HCWs) to full-time contracts, thereby allocating an additional budget for human resources [11]. Austria made a new agreement with increased wages for HCWs [5]. Italy launched an online recruitment drive to establish a Specialist Medical Unit and recruited 300 physicians and 500 nurses to be sent to the most highly affected areas. Each recruited professional received EUR 200 per day in addition to their normal salary as a solidarity premium, which was paid by the Italian Department of Civil Protection [18].

Providing incentives to HCWs: some countries offered reimbursement of health-related expenses or lost revenues attributed to the COVID-19 pandemic, allocated medical insurance, and provided incentive salary and extra paid leave for healthcare workers. For example, USA signed the Coronavirus Aid, Relief and Economic Security (CARES) Act, which provided USD 100 billion to the Department of Health and Human Services to reimburse eligible healthcare providers for healthcare-related expenses or lost revenues attributed to COVID-19 [30]. India allocated medical insurance for healthcare providers during the pandemic [31], which is notable because India's publicly funded insurance programs cover mainly the poor [32]. Brazil, Thailand, and Taiwan reported that they provided unemployment insurance and paid leave for employees who became infected

with COVID-19 [19,31,33]. Latvia, Estonia, and Lithuania raised salaries for frontline workers by 20–100% during the quarantine or for a 3-month period, allowed overtime up to 60 h per week, and introduced a salary bonus for HCWs [20].

The Peruvian government paid extraordinary bonuses to healthcare workers [15] and shortened the length and frequency of shifts for frontline workers without decreasing their salaries [15]. In Italy, to address regional and provincial staffing needs, an online platform was launched to enable HCWs from public hospitals to apply for transfers to the most heavily affected areas, with a financial incentive that included doubling their remuneration [18].

Financial coordination was also undertaken at the hospital level. For example, some Federal States of Germany provided hospitals with financial compensation of EUR 560 per day for unoccupied beds to decrease the bed occupancy rate and thereby enable HCWs to focus on COVID-19-related tasks and COVID-19 patients [11].

Other financial supports were provided to enable non-medical personnel and students to undertake COVID-19-related tasks. For example, EUR 11.3 million were allocated to fund 530 positions in a Containment Scouts Initiative; through this program, more than 500 scouts worked consistently at any one time in 270–380 local health authorities during March to October 2020 on contact tracing, electronic documentation of COVID-19 cases, COVID-19 testing, and supporting telephone services [17]. This program, which provided salaries and contracts for at least 6 months to the containment scouts (including medical students and non-medical personnel), offered longer-term support than other volunteer-based programs in that country [17].

Such financial supports are likely to have motivated HCWs to some extent and were seen to be supportive in dealing with staff shortages. In a long-term disaster setting, however, countries utilizing these short-term strategies may face untenable financial burdens.

### 3.3. Relaxation of Standards and Rules

Some countries relaxed certain quality standards and rules to help decrease the workloads of HCWs and increase workforce numbers. For example, the government of The Netherlands allowed pharmacists to prepare drips at the pharmacy instead of distributing medicines separately [5], which enabled ICU nurses to save time. The Netherlands loosened hiring requirements to support the fast recruitment of HCWs and recruited HCWs whose registration had expired within the prior 2 years [5]. To facilitate and expedite the registration of health professionals, England, Germany, Ireland, and Spain simplified the registration or hiring processes of health professionals [21].

It is not common for doctors and nurses to work longer and back-to-back shifts in Germany, but four states in Germany extended the maximum working hours explicitly to 12 h a day during the COVID-19 pandemic [5]. Cyprus, Greece, Israel, Italy, Malta, Portugal, and Spain also extended hours for the existing staff [18].

### 3.4. Staff Redeployment, Task Shifting, and Skill Mixing

The redeployment and task shifting of HCWs were carried out at international [22,34], national [22,23], subnational [11,22], and institutional levels [24]. For example, at the international level, China sent healthcare workers to Italy at the height of its surge and in its time of need [34].

At the national and subnational levels, workers were redeployed from public or private hospitals to public hospitals in various countries. The Republic of Ireland, Cyprus, Greece, Italy, and Malta took the temporary control of private hospital staff to meet the surge in the needed capacity of the public health system [20,35]. Greece implemented a plan that incorporated transferring patients and staff from public to private hospitals [21]. Some countries required their HCWs to deploy to areas in need: for example, the Thai government mandated that new medical school graduates deploy to rural areas to deal with staff shortages in those areas [19].

At the subnational level, some states of the USA approved temporary rules that allowed nurses licensed in one state to practice in another. This increased the ability of nurses to work across state lines and be mobilized to regions with high healthcare needs [36]. The Governor of New York State requested assistance from HCWs across the USA to support surge responses, particularly in New York City, and this favor was later reciprocated when the state of Utah faced an increased burden of COVID-19 [22]. In Canada, the province of Nova Scotia initiated a "Good Neighbour Protocol" [37] to encourage the "sharing" of HCWs across and within jurisdictions [22].

At the organizational level, the healthcare infrastructure was often reorganized. Many units providing non-essential services were re-purposed to provide services solely dedicated to COVID-19. For example, the doctors and HCWs of these departments who were not directly involved in COVID-19 services were redeployed to areas that needed COVID-19 services [14,22,35,38,39]. Doctors in Italy, particularly general practitioners, modified their practice to provide care primarily through telephone calls or telehealth to enable them to provide services during the COVID-19 pandemic [14].

Task shifting was another option that was often used in dealing with staff shortages. Canada trained more registered nurses to operate ventilators to address the potential for respiratory therapist shortages, given that there were 30 times more registered nurses than respiratory therapists in Canada [22]. This initiative showed that training up more staff who are physically available is effective. In the Rhineland-Palatinate state of Germany, the nursing management was enabled to shift tasks of basic nursing care from nurses to lower-credentialed staff [11].

Redeployment and task shifting had a range of benefits. Overall, redeployment represented an example of the multilateral cooperation and expression of solidarity that appeared as a common-sense response to the COVID-19 pandemic [5,22]. Authorization of mutual support systems gave HCWs greater mobility to work across subnational and state lines to support areas in need. In many countries, emergency legislation paved the way for various approaches aimed at rapidly mobilizing and recruiting health workers [21]. Task shifting, task delegating, and new skill-mixing innovations attained via redeployment leveraged the full scope of skills available within and outside the healthcare workforce [11,22]. Collaborations at the national and subnational levels helped ease hospital overcrowding and provide health services in under-served regions. The voice of solidarity united HCWs and health organizations in aiming together to combat COVID-19 [5]. Peru produced a baseline database to facilitate the identification of skill needs and the design of more appropriate models of redeployment; this country now has a stock of workforce intelligence available on placement choices and role preferences [15]. Some countries relaxed the rules on the number of HCWs required per unit and allowed for more flexible placement of nurses in hospitals [22]. Good leadership helped support successful medical team redeployment and short-term team building: in some cases, leaders allocated human resources based on expertise, which facilitated mutual support, case discussion, and experience sharing. A decentralized leadership approach tended to be better in facilitating targeted training, as local leaders were better able to identify the training needs of redeployees.

At the personnel level, some HCWs perceived redeployment to be a rewarding experience, in that they realized they were fulfilling a much-needed role for patients [5]. Both physicians and nurses felt their professional value grew by supporting COVID-19 treatment [23,40]. They said they were recognized for their own value and could put their knowledge and skills into practice. They thought that having the chance to go to Wuhan to support the COVID-19 outbreak was a commendable opportunity [25]. In addition, reallocating duties away from staff in short supply enabled ICU nurses and doctors to focus on their specialist non-transferable skills. It also enabled exchanges and mutual help between medical teams, such as through sharing of protective materials, case discussions, and exchange of treatment experience [41,42].

A key barrier for successful redeployment planning was difficulty in measuring human resource needs, such as which specific roles were in demand and who was available and

healthy [43]. The kind of needs assessment that would fill this gap was also found to be crucially required at the organizational level during external and internal redeployment. Peru reported on efforts to develop a baseline dataset that could be used in identifying skill gaps, creating more suitable redeployment models, and offering a repository of information on the skills, preferred roles, and placements of workers [15].

Jurisdictions often lacked some basic guidelines on workforce adequacy, such as the required staffing ratios of physicians and nurses during "normal" and "emergency" periods, hindering efforts to monitor and adjust resources during the pandemic [22].

Surges of infected HCWs, which led to staffing shortfalls in some institutions, was another unanticipated problem for workforce mobilization teams [25,43].

If the requisite skill set was not already present in the worker cadre, new recruits needed to be upskilled to support COVID-19-related services. The Nightingale Hospital in London, UK reported that their redeployed workforce was not specifically trained on ICU services [26]. Therefore, some countries emphasized that it is crucial to minimize training needs and maximize the use of HCWs' experience and current knowledge by placing them in duties where their existing skills can be more easily transferred [27,43–45].

Since redeployment during COVID-19 was quite a sudden decision for many hospitals, across all specialties and grades, both private and public, it brought swift and significant disruptions to doctors' working lives [35]. The redeployed teams were made of staff from different hospitals and departments, most of whom were strangers to one another [41]. Often, they were not ready to shift to new sites and schedules, especially when involving night and weekend shifts [43]. Some struggled with redeployment and/or were unable to engage in their stated roles [24] because of high callout rates during their shifts. Redeployed HCWs reported great concern regarding professional loss (such as loss of educational opportunities in their chosen profession) [43], as well as intense fear of COVID-19 infection [25,43]. Redeployment of those with less experience in inpatient care to inpatient areas was a particular cause of fear [43]. This was further exacerbated by the risk or experience of psychological consequences, such as anxiety and depression [23].

Boosting the appropriate health workforce was a crucial prerequisite for increasing the capacity of ICU beds, which particularly require adequate qualified staff [18]. Overall, the analyzed reports indicate that redeployment during a public health emergency requires sound workforce management. An internal incident management team and a panel of external experts could be leveraged to provide the necessary tools and guidelines for planning, monitoring, and managing the health workforce during a prolonged and/or rapidly changing pandemic [16].

### 3.5. Recruiting Volunteers/Fast-Tracking Medical Students

The pool of healthcare workers from which a national hospital could recruit was small, even in developed countries [26]. Therefore, some countries used volunteers and students to cope with staff shortfalls. Many volunteers from government, clinical, public health, and non-medical areas were recruited to deal with COVID-19 surges [11,16,18,28,46–50]. For example, Malaysia utilized volunteers from non-healthcare ministries, government agencies, and NGOs. These collaborations resulted in a seamless distribution of the workforce throughout Malaysia [23].

In some countries, medical-licensed volunteers were upskilled to support doctors and nurses in ICU units [26,51,52]. For example, podiatrists, optometrists, and school nurses in the UK were authorized to change pre-prepared infusions, maintain patient hygiene, and assess pressure areas after being upskilled at Nightingale Hospital London [26]. Similarly, to support ICU patients, point-of-care ultrasonography was performed by radiologists [26].

Non-medical-licensed volunteers were also leveraged to support the tasks that did not require medical skills. In Nightingale Hospital London, non-medical volunteers whose routine responsibilities were suspended due to COVID-19, such as academics pursuing postponed non-COVID-19 research and flight attendants whose flights were deferred, were recruited and trained to record observations of COVID-19 cases [26].

In Malta, tourism and business sector workers whose regular jobs had been curtailed were used as volunteers for non-medical roles, such as on contact tracing teams [18].

Volunteerism among medical and nursing students was relatively common [28]. The methods used to recruit medical student volunteers varied by country. Some countries sent calls for volunteers [28] while others had existing recruitment systems and thus could access lists of students available for volunteer activities during an emergency [29].

The medical student volunteers supported HCWs by performing short-term functions in healthcare systems, especially in public health [16,47], such as by working telephone lines for COVID-19-related communication with patients [16], planning PPE distributions and patient communication, and engaging in outreach programs to geriatric patients [28]. In The Netherlands, medical students supported general practices and provided health information to the general population [51]. Fast-tracking trainees near the end of their programs was another common strategy [15]. In Ireland, medical school examinations were advanced to speed up the entry of students to the healthcare workforce, bringing approximately 1000 intern doctors into the workforce ahead of the normal schedule [53]. In Australia, nursing students were employed as assistant nurses, enabling registered nurses to handle more acute cases. In Germany, more than 20,000 medical students signed up to participate in clinical practice in response to a call by the federal medical student association [22]. Cyprus, Israel, Italy, Portugal, and Spain loosened hiring requirements to enable the rapid recruitment of additional staff on short-term, freelance, or temporary contracts. For example, Portugal used an exceptional procedure to hire an extra 137 doctors and 1100 nurses by the end of July 2020 [18].

The most common challenge in recruiting students was the need for mechanisms to identify available students, call them, and manage them. In the absence of a medical volunteer registry, significant effort was required to screen an overwhelming number of volunteers and call the selected individuals. Icahn School of Medicine at Mount Sinai became an example for good management and leadership of volunteer activity: the school created a Student Council Emergency Preparedness Committee to build a student labor force from third- and fourth-year medical students, graduate students, and nursing candidates. The committee distributed the volunteers into task groups, such as pharmacy, administrative services, hospital operations, labs/research, personal protective equipment, telehealth, and morale. Once a detailed request was received from a hospital department, the head of each task group would assign the available students to the field [29].

*3.6. Using Other Workforce Resources*

Other workforce resources used by countries in responding to the pandemic included additional contract workers, inactive healthcare workers, returnees whose registration had lapsed within the prior 1–2 years, and internationally educated health professionals [11,15].

Recalling inactive healthcare workers, such as retired staff [11,22] and/or those on leave, was a common response across many countries. In The Netherlands, for example, 20,000 retired or on-leave health workers expressed their willingness to come back to the health sector in response to COVID-19 [22]. In Germany, the president of the German Federal Association called upon retired physicians to return and help with some public health tasks, such as tracing cases and manning telephone helplines [22,54]. Regulatory authorities responded quickly to enable the inactive practitioners to come back to work [55]. Italy and Portugal also recruited retired or inactive HCWs [18].

The inactive health workforce hugely contributed to the response to the COVID-19 pandemic, but the decision to call for such forces carried some concerns, including the often-neglected need for short-term training and the greater potential risks posed to older workers by the virus [18,22,56]. In many cases, it would have been helpful to have an official reserve list of available health professionals. An NGO set up a database of inactive health workers who could volunteer at a time of need, and hospitals worked together to reassign staff with COVID-19 training [20].

Integrating internationally educated health professionals was another strategy utilized by some countries. In the UK, an accelerated process was implemented to quickly place international nurses onto a temporary register that was slated to close on 30 September 2022 [57]. In Ireland, almost 400 retired or overseas workforce members offered their help in meeting the surge capacity in Ireland [35]. Some Indigenous communities of Canada proposed to use Cuban doctors to support the response to COVID-19 [58]. England extended visas for frontline workers from abroad, enabling them to continue working.

Moreover, additional contract workers were utilized in different settings, such as on Rapid Response Teams, which were responsible for clinical assessment support, COVID-19 testing, sampling, and tracing; Clinical Monitoring Teams at care-monitoring centers and isolation units; and Humanitarian Corpse Collection Teams, which supported the care of Indigenous people [15]. This kind of deployment reflects on the use of a flexible strategy in responding to initial containment needs and shifting to community- and hospital-based service needs as the infection spreads throughout the country. However, maintaining this level of service presents budgetary issues.

In addition, some countries including Greece, Israel, Italy, and Spain temporarily enrolled professionals from the armed forces for the COVID-19 response [18].

While the above-described strategies were supported by policy/decision-makers of some countries [58], they were opposed by others [59]. For example, Canadian local politicians were opposed to inviting foreign-trained doctors and nurses due to safety implications [59]. Mexican politicians were concerned that these strategies neglected endemic local underemployment [60]. Additional concerns included whether the additional personnel would be able to properly perform the tasks, and how to legalize their contracts over the course of the epidemic [22]. That said, non-US citizens employed by states are required to adhere to US visa and guest-worker restrictions, and thus are required to leave their posts at periodic intervals in the absence of specialized waivers [61].

## 4. Discussion

The current review identified several types of strategies that were used or proposed as ways to address human resource shortfalls during COVID-19. None were perfect; all such strategies have pros and cons that should be weighed and adjusted for in a given context. The main strategies identified in this review include using financial mechanisms to support new staff hiring, providing incentives, relaxing the quality standards and rules, engaging in redeployment, calling up inactive HCWs, recruiting volunteers and students, and utilizing internationally educated healthcare workers. These strategies are in line with the recommendations made by the WHO in their COVID-19 guidance for human resources, which was promulgated to aid health managers and policy makers in managing the human resource problems encountered during the COVID-19 pandemic [62].

Our findings are also consistent with those captured by the Health System Response Monitor published by WHO, identifying the pandemic responses of European countries and some Asian countries [9,10]. However, these reports did not discuss any advantages, disadvantages, best practices, and lessons learned for the strategies implemented in the assessed countries. Therefore, in addition to broadening the analysis of implemented strategies, we investigated the advantages, disadvantages, best practices, and lessons learned from the strategies identified in our review, and wherever available the impact of COVID-19 on human resource management in the post-COVID-19 era.

We found that redeployment of HCWs was one of the most commonly utilized strategies. We believe that, at all levels, this strategy exemplified solidarity in combating the pandemic. Despite its positive features, however, this strategy carried several challenges at the personnel, organization, and possibly system levels. The personnel-level challenges, which included stress due to sudden schedule changes, anxiety about professional losses, and fear of exposure to infection, may be managed by utilizing some of the best practices identified in our review. In general, the well-being of all deployed personnel must be

tracked through continuous guidance and close supervision; this would be enabled by monitoring each HCW's occupational health and safety within healthcare facilities.

Most importantly, close supervision and support are needed to avoid the risk of HCWs performing poorly or improperly and thereby decreasing patient safety. For example, the Australian College of Critical Care Nurses, Ltd., suggested that any registered nurse who lacks critical care nursing experience and is asked to deliver patient care should be adequately supported and supervised by a critical care nurse, with the goal of optimizing patient safety and outcomes [63]. The same group stated that the provision of direct patient care by a nursing student or health professional other than a registered nurse should be implemented only in extreme situations [63].

Personnel- and organization-level strategies are important, but system-level coordination is needed to support sustainability. For example, the development of a centralized system that would help identify expertise available for deployment may be crucial in human resource mobilization management [23]. Ridley et al. also mentioned the importance of clarifying the jurisdiction responsible for identifying the staffing ratio that is to be followed during redeployment [64]. As a workaround, Australia used previously established staffing ratios to estimate the numbers of additional ICU nurses and physicians that would be required to operationalize additional ICU beds and ventilators during the COVID-19 outbreak [65].

The recruitment of volunteers was another common strategy. The use of volunteers was expected to alleviate the workload of HCWs during the pandemic, especially in the public health sector and/or when good coordination mechanisms were available. Waitzberg et al. reported that there was a staff shortage in the public health sector, requiring all HCWs to initially put forth maximum effort in quarantine practices, tracing, etc., and that this led to a high rate of infection among HCWs. However, later strategies involved using volunteers to manage these public health tasks [18]. Again, this kind of strategy (i.e., using volunteers) might be more sustainable given access to a centralized approach, such as a volunteer database; such a database could be similar to a blood donor database, but contain information on the skills and preferences of volunteers available during an emergency. An online recruitment system could be another option, as it would allow volunteers to register at any time by providing key information, such as their skills, roles, preferences, etc. Going forward, careful data management and relevant information availability might be keys to overcoming some of the challenges faced in disaster management, such as those seen during the COVID-19 pandemic [15,23].

This kind of data management system could also be useful for recalling inactive workers. For example, an official reserve list of available health professionals could be helpful. In Estonia, an NGO set up a database of inactive healthcare workers who could volunteer in case of need, and hospitals worked together to reassign staff with COVID-19 training [20]. Marshall AP et al. suggested that early identification of recallable HCWs with critical care experience would be helpful [66].

Internationally educated health professionals were integrated in some developed countries [35,67]. This strategy may be more feasible in developed countries that receive immigrants who are licensed nurses, or in countries that can manage such responses financially. It is likely to be infeasible in developing countries.

Financial support from governments was found to motivate HCWs. However, such efforts may have some drawbacks. One-time or short-term incentives may not be effective at keeping healthcare workers in their jobs, especially if the working conditions are chronically understaffed [68]. Financial incentives may not address the root cause of the healthcare workforce shortage [68,69]. It also may not be sustainable in the long term, especially if the healthcare system is underfunded or if the incentives are not budgeted for in the long term [68]. Financial incentives may not be equitable, as they may only be available to certain healthcare workers or in certain regions, leaving others without the same benefits [70]. Although financial incentives can motivate the HCWs to some extent, their other needs may

not be addressed properly, such as adequate personal protective equipment, COVID-19 testing, proper training, safe staffing, and mental health care [69].

Relaxing the quality standards and rules for fast recruitment of HCWs was another option in the support of the healthcare workforce shortage. However, it may lead to potential risks and challenges. Healthcare professionals are already at high risk of being infected with COVID-19 due to their direct contact with patients [71]. Relaxing recruitment standards and rules may lead to the hiring of less qualified or inexperienced staff, which could increase the risk of infection for both healthcare professionals and patients. Hiring staff without proper training could lead to inadequate care and increase the risk of medical errors [72]. This could also lead to burnout and stress among healthcare professionals who may have to take on additional responsibilities to compensate for the lack of trained staff [73]. Healthcare professionals are facing significant mental health challenges during the pandemic, including anxiety, depression, and burnout [71]. Thus, hiring less qualified or inexperienced staff could increase the workload and stress levels of existing healthcare professionals, which could exacerbate mental health concerns. Some countries allowed HCWs for extended hours. It also may cause potential risks such as burnout and fatigue of HCWs [74] and disruption of work–life balance [74] which in turn may lead to decreased quality of care and increased medical errors [74].

This review has encompassed articles that examined diverse responses to address healthcare workforce shortages in various contexts, including intensive care units, internal medicine units, and both urban and rural health facilities in 50 countries. These articles offered valuable insights into the challenges and potential solutions related to healthcare workforce shortages. However, it is important to note that the findings may not be universally applicable to all nations or contexts. Therefore, the current findings should be considered as potential options for addressing the issue rather than definitive conclusions that can be applied universally.

As it was mentioned and addressed in the "Results" section, we identified several best practices and lessons learned that can be reflected in a future pandemic. That said, we did not identify any sustainable change in the human resource function of Health-EDRM systems. It is possible that such changes have been experienced/detected since the publication dates of the reviewed papers. To address this gap, we have generated case studies focusing on Iran [75], Italy [76], Japan [77], Korea [78], and the USA [79], with the goals of identifying lessons learned, the best practices of the strategies used by each country during the COVID-19 pandemic, and their impact on Health-EDRM systems with emphasis on human resource management and health service delivery. These case studies have been published as a part of this project. Based on the findings of our review and case studies, we strongly encourage countries to publish their experiences with COVID-19 responses, sharing the best practices and lessons learned. Such information would be particularly useful from low- or low-middle income countries, where the healthcare workforce systems were at high risk of being affected by COVID-19, but reports are scarce. Future research needs to investigate how these countries dealt with human resource challenges and experiences, so that this information can be shared with other countries with similar resource levels.

To date, countries studied herein have not determined clear policies on how to ensure the sustainability and resilience of the healthcare workforce, particularly during major health system shocks. Therefore, a follow-up study of the implemented strategies is needed to investigate their effects, best practices, lessons learned, and related system changes. Best practices should be shared, because countries with experience in responding to SARS and MERS also responded well during the COVID-19 outbreak [80].

The current review has some limitations worth mentioning. First, the review covered only the academic literature because searching for gray literature is time- and resource intensive, as it often involves searching multiple sources and using different search strategies [81,82] and they may be published in local languages, so the current review may have provided a limited coverage and instead fostered a balanced picture of available evidence. Hence, the current findings may have not fully captured all utilized strategies and their best

practices and lessons learned. However, some of the reviewed papers included numerous gray literature sources and policy reports published in local languages. Second, we selected only English-language articles, so our review would have missed non-English language studies. Third, it is important to note that the period of review for our scoping review spans from 2020 to August 2022. As a result, it is essential to acknowledge that the most recent developments and information beyond August 2022 have not been captured in our analysis. It is crucial to recognize that the field of healthcare workforce management is dynamic and constantly evolving and new studies, policies, and initiatives may have emerged since the conclusion of our review, and thus, the most up-to-date information may not be fully reflected in our findings. Fourth, most of the included papers were from high-income- or upper middle-income countries, limiting our ability to investigate strategies implemented in low-resource settings. Even the relevant publications of the WHO lack information on developing or less-developed countries, where more information is needed on best practices and lessons learned. Growth in the demand for healthcare workers will be highest among upper middle-income countries, driven by economic and population growth and aging [83]. Middle-income countries will face workforce shortages because their demand will exceed supply [83]. Low-income countries will face low growth in both demand and supply, which are estimated to be far below what will be needed to achieve an adequate coverage of essential health services [83]. Therefore, while country-level studies can identify specific needs and challenges of each country, regional-level studies are also important to provide a broader perspective on the global shortage of health workers and its impact on different income groups.

## 5. Conclusions

The current review identified common strategies that 50 countries implemented to address the lack of healthcare workers during the COVID-19 pandemic, as well as their best practices and lessons learned. The information in this review does not represent an exhaustive list of the countries that used the identified strategies. All of the strategies discussed herein had advantages and drawbacks; none were ideal. Thus, in the event of the next pandemic or disaster, country-level responses must be implemented with consideration of the country's specific situation. This review focused primarily on developed countries—even the WHO's reports and publications are mostly available for only developed countries, along with some European developing countries—so international organizations could mobilize to help developing or less developed countries by sharing their best practices and lessons learned.

**Supplementary Materials:** The following supporting information can be downloaded at: https://www.mdpi.com/article/10.3390/su151511668/s1, Table S1: Detailed information extracted from the reviewed articles.

**Author Contributions:** Conceptualization, O.C.-O. and T.K; methodology, O.C.-O.; software, C.O; validation, O.C.-O., J.A. and T.K.; formal analysis, O.C.-O.; investigation, O.C.-O. and J.A.; data curation, O.C.-O. and J.A.; writing—original draft preparation, O.C.-O.; writing—review and editing, N.G., J.A., Y.Y., R.K., F.V.T., V.M. and T.K.; visualization, O.C.-O.; supervision, O.C.-O. and T.K.; project administration, O.C.-O.; funding acquisition, O.C.-O. All authors have read and agreed to the published version of the manuscript.

**Funding:** This study was funded by the World Health Organization Kobe Centre for Health Development (WKC-HEDRM-K21001).

**Institutional Review Board Statement:** Not applicable.

**Informed Consent Statement:** Not applicable.

**Data Availability Statement:** Detailed information on reviewed articles in Supplementary Table S1 and all articles can be found in either of four databases used in Medline, Web of Science, CINAHL, and TRIP.

**Conflicts of Interest:** Kayano Ryoma works at the funding organization, but the funder had no role in the design of the study; in the collection, analyses, or interpretation of data; in the writing of the manuscript; or in the decision to publish the results.

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
