# Peer review of "Impact of COVID-19 on Health Emergency and Disaster Risk Management System: A Scoping Review of Healthcare Workforce Management in COVID-19"

_sustainability, doi:10.3390/su151511668_

Round 1
Reviewer 1 Report
Please the document, there is minor revisions.

Author Response
Thank you very much for your valuable time for reviewing our work.
The final sentence in the abstract has been removed as English editing company suggested that “Final sentence of abstract is quite vague and does not fit well with the text above. Can it be removed?”.

Reviewer 2 Report
Introduction
· The introduction provides a clear and concise overview of the global impact of the COVID-19 pandemic, emphasizing its disruptive effect on the health sector's human resources and infrastructure.
· The inclusion of the WHO's Health Emergency and Disaster Risk Management (Health-EDRM) Framework adds a relevant and authoritative context to the study.
· The identification of gaps in existing reports and the aim to provide comprehensive information on country-level responses, advantages, disadvantages, best practices, and lessons learned is commendable.
· The introduction lacks specific information on the scope and methodology of the scoping review, making it difficult to assess the study's rigor and potential biases.
· The reference to "our knowledge" regarding WHO collaborations and the absence of specific sources or citations raises concerns about the reliability and accuracy of the information presented.
· The focus on country-level responses may overlook important regional or local variations in workforce shortage strategies, limiting the generalizability of the findings.
· The introduction could benefit from providing a clear research question or objective to guide the scoping review.
Methods
· The materials and methods section provides a clear description of the scoping review framework based on the Joanna Briggs Institute (JBI) Framework of evidence synthesis.
· The use of the PCC (Participant, Concept, Context) framework to define the research questions ensures a systematic approach to identifying relevant information.
· The inclusion and description of the search strategy and key terms used in searching the electronic databases enhance the transparency and reproducibility of the review process.
· The use of two researchers for screening and a third researcher as an adjudicator helps ensure consistency and minimize bias in the selection of relevant studies.
· The presentation of data extraction items in an Excel spreadsheet demonstrates a structured and organized approach to collating the results.
· The materials and methods section lacks details on the inclusion and exclusion criteria for the study selection process, such as specific criteria for determining relevance or quality assessment.
· The timeframe for literature inclusion (January 2020 to August 2022) may limit the inclusion of more recent studies that could provide valuable insights into evolving strategies.
· The restriction to English-language publications may introduce language bias and exclude relevant studies published in other languages.
· The absence of a quality assessment or risk of bias evaluation for the included studies limits the ability to gauge the reliability and validity of the evidence synthesized.
· The description of the qualitative narrative synthesis is brief, and it would be beneficial to provide more details on the analytical approach and methods used.
· Overall, the materials and methods section provides a solid foundation for conducting the scoping review. However, to enhance the transparency and rigor of the study, it is important to address the limitations mentioned, provide clearer details on the inclusion criteria, and consider incorporating a quality assessment component to evaluate the included studies. Additionally, expanding on the methods used for the qualitative narrative synthesis would strengthen the overall methodology.
Results
The study utilized a bibliometric analysis to gather a comprehensive set of articles, which enhances the credibility of the findings.
The inclusion of relevant references cited in the analyzed papers ensures a thorough examination of the topic.
The study provides a clear breakdown of the types of papers analyzed, including commentaries, reviews, special reports, and research articles, which allows for a better understanding of the literature landscape.
The number of articles analyzed (18) may be considered relatively small, which could limit the generalizability of the findings.
The absence of articles covering the year 2022 raises questions about the comprehensiveness and up-to-datedness of the analysis.
The reliance on databases and the exclusion of other potential sources of literature may introduce selection bias and limit the scope of the study.
The article lacks a detailed description of the methodology employed for the bibliometric analysis, such as the search terms used and the criteria for article selection. Providing this information would enhance the transparency and replicability of the study.
The distribution of the analyzed papers by publication year (61% from 2020, 39% from 2021) is not adequately justified or discussed. Clarifying the rationale behind this distribution would strengthen the study.
The study could benefit from a more critical discussion of the strengths and weaknesses of the included articles. Evaluating the quality of the literature would provide insights into the reliability and validity of the analyzed information.
The study provides a comprehensive overview of different financial coordination mechanisms adopted by various countries during the COVID-19 pandemic.
The inclusion of specific examples and financial figures adds depth to the analysis and facilitates understanding.
The study highlights the positive impact of financial coordination mechanisms, such as hiring new staff and providing incentives to healthcare workers, in addressing staff shortages and motivating healthcare professionals.
The focus on a limited number of countries may limit the generalizability of the findings and overlook potential strategies implemented by other nations.
The absence of a comparative analysis or discussion of potential drawbacks and unintended consequences of the financial coordination mechanisms reduces the overall depth of the study.
The article does not discuss the long-term sustainability and feasibility of the financial supports provided. Considering the potential financial burdens in the future would strengthen the analysis.
The article lacks clarity regarding the source of information and data used to support the claims made about the financial coordination mechanisms. Providing references or citations would enhance the credibility of the study.
The absence of a comprehensive analysis of the cost-effectiveness and efficiency of the implemented financial strategies limits the evaluation of their overall impact.
The study would benefit from a more nuanced discussion of the potential equity implications of the financial supports provided to healthcare workers, such as the distribution of resources and the inclusion of marginalized groups.
The study provides a detailed account of the relaxation of standards and rules to alleviate the workload of healthcare workers during the COVID-19 pandemic.
The inclusion of examples from multiple countries demonstrates the global nature of the issue and the diversity of approaches taken.
The discussion of the benefits of staff redeployment, task shifting, and skill mixing highlights the importance of flexibility and adaptability in healthcare systems.
The article lacks an in-depth exploration of the potential risks and challenges associated with the relaxation of standards and rules. Understanding the trade-offs involved would provide a more balanced analysis.
The absence of data or empirical evidence supporting the claims made about the benefits and outcomes of staff redeployment and task shifting limits the strength of the conclusions drawn.
The study does not address the long-term implications of the implemented strategies or discuss the sustainability
Discussion
Comprehensive coverage: The discussion section provides a comprehensive overview of various strategies identified to address human resource shortfalls during COVID-19. It includes a range of strategies such as financial mechanisms, incentives, relaxation of quality standards, redeployment, recruitment of volunteers and students, and utilization of internationally educated healthcare workers.
Alignment with WHO guidance: The strategies identified in this review are in line with the recommendations provided by the World Health Organization (WHO) in their COVID-19 guidance for human resources. This strengthens the validity and relevance of the findings.
Consistency with other sources: The review findings align with the Health System Response Monitor published by WHO, which further enhances the credibility of the identified strategies.
Limited scope of literature review: The review primarily focuses on academic literature, which may have resulted in the exclusion of relevant strategies, best practices, and lessons learned from non-academic sources. This limitation suggests that the findings may not fully capture the complete range of utilized strategies and their implications.
Language bias: The review only includes English-language articles, which may have led to the omission of non-English language studies. Consequently, important strategies implemented in other languages may have been overlooked, limiting the comprehensiveness of the analysis.
Geographic bias: The majority of included papers are from high-income or upper middle-income countries, which restricts the ability to thoroughly investigate strategies implemented in low-resource settings. Consequently, the generalizability of the findings to less-developed countries may be limited.
Lack of critical analysis: The discussion section primarily focuses on describing the strategies and their alignment with WHO recommendations, without critically evaluating their effectiveness or efficiency. Providing a more critical analysis of the pros and cons of each strategy would enhance the depth of the discussion and provide more practical insights for health managers and policymakers.
Incomplete information on impact: The review mentions the identification of advantages, disadvantages, best practices, and lessons learned for the strategies, but it does not provide specific details or examples of these findings. Including more concrete information on the impact and outcomes of the strategies would strengthen the practical applicability of the review.
The English writing style of this manuscript is clear and concise, making it easy to follow and understand the information presented. The sentences are well-structured, and the ideas flow logically from one paragraph to the next. The use of appropriate vocabulary and terminology related to healthcare worker shortages and pandemic response adds credibility to the text.
However, there are a few areas where the writing style could be improved. Firstly, there are instances where the author could have used more varied sentence structures to enhance the overall readability of the manuscript. Additionally, some sentences could benefit from being broken down into smaller, more digestible chunks to avoid potential confusion.
Furthermore, while the manuscript effectively presents the identified strategies and their advantages and drawbacks, there is room for more engaging and dynamic language to capture the reader's attention. Injecting a sense of enthusiasm and conviction in the writing can help make the conclusions and recommendations more impactful.
Overall, the manuscript demonstrates a solid English writing style, but with some refinement, it could further enhance the clarity and engagement of the text.
Author Response
Thank you for your valuable time, thorough review and thought-provoking comments. Your comments definitely helped with the improvement of the paper. We have reflected on the comments in the revised version of the manuscript, unless otherwise specified. The modifications are shown in red text. Please kindly find the response and response letter in attachment.

Round 2
Reviewer 2 Report
The authors did an excellent job in reviewing their manuscript.
minor spelling check.